# Quality of Life Scores Remained Different among the Genotypic Groups of Patients with Suspected Hemochromatosis, Even after Treatment Period

**DOI:** 10.3390/genes13010118

**Published:** 2022-01-10

**Authors:** Luis Alfredo Utria Acevedo, Aline Morgan Alvarenga, Paula Fernanda Silva Fonseca, Nathália Kozikas da Silva, Rodolfo Delfini Cançado, Flavio Augusto Naoum, Carla Luana Dinardo, Alexandre Costa Pereira, Pierre Brissot, Paulo Caleb Junior Lima Santos

**Affiliations:** 1Department of Pharmacology—Escola Paulista de Medicina, Universidade Federal de Sao Paulo (EPM-Unifesp), Sao Paulo 05403-904, Brazil; luisalfredoutria@hotmail.com (L.A.U.A.); a.cmorgan@hotmail.com (A.M.A.); nathalia.kozikas@unifesp.br (N.K.d.S.); 2Laboratory of Genetics and Molecular Cardiology, Heart Institute (InCor), University of Sao Paulo Medical School, Sao Paulo 05403-904, Brazil; paulafonseca@usp.br (P.F.S.F.); acplbmpereira@gmail.com (A.C.P.); 3Hematology and Hemotherapy Section, Santa Casa Medical School, Sao Paulo 05403-904, Brazil; rdcan@uol.com.br; 4Academia de Ciência e Tecnologia, Sao Jose do Rio Preto 15020-130, Brazil; fnaoum@hotmail.com; 5Fundação Pró-Sangue, Hemocentro de São Paulo, São Paulo, Brazil, Universidade de São Paulo (USP), Sao Paulo 05403-904, Brazil; caludinardo@gmail.com; 6Institut NuMeCan, Inserm U-1241, Univ Rennes 1, 35000 Rennes, France; pierre.brissot@gmail.com

**Keywords:** hemochromatosis, short form health survey, quality of life, SF-36, phlebotomy, iron overload

## Abstract

Background: Hemochromatosis is a genetic condition of iron overload caused by deficiency of hepcidin. In a previous stage of this study, patients with suspected hemochromatosis had their quality of life (QL) measured. We observed that QL scores differed among genotypic groups of patients. In this reported final phase of the study, the aims were to compare QL scores after a treatment period of approximately 3 years and to analyze a possible association of the serum ferritin values with QL scores. Methods: Sixty-five patients were enrolled in this final phase and divided into group 1 (patients that showed primary iron overload and homozygous genotype for the *HFE* p.Cys282Tyr mutation) and group 2 (other kinds of genotypes). Short Form 36 (SF-36) was performed and consisted of eight domains with a physical and also a mental component. Results: Both groups had a significant decrease in serum ferritin concentrations: group 1 had a variation from 1844 ± 1313 ng/mL to 281 ± 294 ng/mL, and group 2 had a variation from 1216 ± 631 ng/mL to 236 ± 174 ng/mL. Group 1 had a smaller mean value for these six SF-36 domains compared with group 2, indicating a worse QL. Conclusions: In this final stage, six domains demonstrated a difference among genotypic groups (role emotional and mental health, adding to the four of the initial phase), reassuring the impact of the identified genotype on the QL of hemochromatosis patients. Furthermore, despite that both patient groups demonstrated similar and significant decreases in serum ferritin values, no association was found between the decrease in this biological parameter and the SF-36 domains.

## 1. Introduction

Hemochromatosis is a genetic condition of iron overload caused by deficiency of hepcidin, making the absorption of dietary iron exceed the requirements. Hemochromatosis excludes the acquired iron overload [1,2], and it is caused by mutations in the five main genes; however, it is more frequently associated with the p.Cys282Tyr (C282Y) homozygous genotype, which is the most common genetic alteration associated with hemochromatosis, in the *HFE* gene [1,3]. The diagnosis is based on laboratory tests (transferrin saturation and serum ferritin) [4], genetic tests, and/or magnetic resonance imaging (which helps with the quantification of iron). Regarding treatment, phlebotomy is the safest and cheapest option to remove iron excess in the body in order to reduce mortality and comorbidities [1,3,5,6,7,8].

The World Health Organization (WHO) defines quality of life (QL) as “the individuals’ perception of their position in life in the context of the culture and value systems in which they live and in relation to their goals, expectations, standards and concerns” [9,10]. The Short Form 36 (SF-36) questionnaire is a popular generic instrument to evaluate health-related quality of life [11]. It consists of eight domains comprising physical and mental component scores, and it has been validated for the Brazilian population in the Portuguese language [12,13].

In this scenario, the main aims of the reported final phase of this study were to compare QL scores according to genotypic groups of patients after a treatment period of approximately 3 years and to further evaluate a possible association of the serum ferritin values with QL scores.

## 2. Methods

### 2.1. Patients

In an initial phase of the study by our group, evaluating the QL of 79 patients with suspected hemochromatosis, we observed that four of the eight domains of the SF-36 questionnaire were significantly different among genotypic groups of patients. Group 1 (patients with primary iron overload and that demonstrated homozygous genotype for the *HFE* p.Cys282Tyr) had worse QL scores compared with group 2 (the one that had the patients with primary iron overload and other genotypes) [13]. In this scenario, the main aims of the reported final phase of this study were to compare QL scores according to the genotypic groups of patients after a treatment period of approximately 3 years and to further analyze a possible association of the serum ferritin values with QL scores.

The study protocol was approved by the research ethics committee of the Universidade Federal de São Paulo CEP/Unifesp (1195/2018), and consent forms of the patients were obtained prior to entering the study. Patients were selected from hematologic clinics (Ambulatório do Hemocentro da Santa Casa—São Paulo, Ambulatório de Hematologia do Hospital das Clínicas—São Paulo, and Instituto Naoum de Hematologia—São José do Rio Preto, Brazil). The inclusion criteria were: age older than or equal to 18 years, transferrin saturation (TS) greater than or equal to 45%, and serum ferritin (SF) ≥300 ng/mL for males or ≥200 ng/mL for females. Otherwise, the exclusion criteria were: patients with positive serology for hepatitis (B or C), alcoholic liver disease, high alcoholic consumption (more than 20 g daily), hemolytic anemias, repeated blood transfusions, metabolic syndrome, or insulin resistance not resulting from hemochromatosis. In this study, we also excluded patients with juvenile hemochromatosis carrying *HAMP* and *HJV* mutations [14,15].

The first phase of this study was described by Fonseca et al. [13]. In this phase, 79 patients were analyzed, in which they proposed to verify whether QL domains, assessed by SF-36, were different or not according to genotypic groups in the patients with suspected hemochromatosis. Two meetings were held with these patients in this step. Between these procedures, the patients were followed up by their doctors of origin. After this analysis, the patients received a report and guidance on their results. The second part of this study aimed to compare the QL scores after a treatment period of approximately 3 years in order to assess a possible association of serum ferritin values with QL scores. It is important to consider that we had a change in the total number of patients. Thirteen patients were lost to follow-up, and one patient died. Considering this, we were able to enroll 65 patients (after the initial treatment period of approximately 3 years). In this second phase of the study, calls were made, e-mails were sent, and letters and remote conversations were obtained to gather information. During these 3 years, the patients were followed up by their doctors of origin, and they were randomly included. Patients who were able to participate in the study generally accepted. We do not have data on patients who agreed to participate and those who declined. All who agreed to participate signed the informed consent form.

### 2.2. SF-36 Health Survey

The SF-36 questionnaire consists of 36 questions and measures eight domains: physical functioning, role physical, bodily pain, general health, vitality, social functioning, role emotional, and mental health. All participants in groups 1 and 2 answered the SF-36 questionnaire applied by an interviewer, or it was self-administered and reviewed by an interviewer. Each of the 36 questions has a fixed value. After, these values were converted to raw scale, which provided the score of each domain. The values range from 0 to 100 points, where higher scores indicate a better health condition [11,16,17,18].

### 2.3. Statistical Analysis

Categorical variables are represented as percentages, while continuous variables are represented as means ± standard deviations. Kolmogorov–Smirnov test was used to test normality. Chi-squared or Fisher tests were performed to compare the categorical variables (such as gender, self-declared race/color, education level, and consumption of alcohol) according to the genotypic group. Student’s *t*-test was used to compare the values of the SF-36 domains and serum ferritin between groups 1 and 2. The values of the SF-36 domains and serum ferritin were adjusted for age and gender. SF-36 domains and serum ferritin values also had paired analyses (paired-samples Student’s *t*-test) comparing initial and final phases. Pearson correlation test was performed for the variables’ initial and final serum ferritin values and SF-36 domains. The level of significance was defined at *p* ≤ 0.05. All statistical analyses were carried out using the SPSS software (Version 20.0).

## 3. Results

Sixty-five of 79 patients were enrolled in this final phase of the present study. Two genotypic groups were formed: group 1 consisted of patients with primary iron overload and homozygous genotype for the *HFE* p.Cys282Tyr mutation (*n* = 23), and group 2 consisted of patients with primary iron overload and other genotypes (compound heterozygosity for the p.Cys282Tyr/p.His63Asp mutation (*n* = 10), heterozygosity for the p.Cys282Tyr (*n* = 4), homozygosity (*n* = 10), heterozygosity (*n* = 8) for the p.His63Asp, or absence of p.Cys282Tyr or p.His63Asp (*n* = 10)). There was a loss of treatment follow-up for six and eight patients from groups 1 and 2, respectively.

Table 1 shows the general characteristics related to genotypic groups of the patients. Group 1 had a lower percentage of males (34.8%) and similar mean serum ferritin values (281 ± 294 ng/mL) compared with group 2 (85.7%, 236 ± 174 ng/mL; *p* = 0.001, *p* = 0.53, respectively).

Figure 1 shows comparisons of mean serum ferritin values of groups 1 and 2 according to the initial and final phases. Group 1 had mean serum ferritin values of 1844 ± 1313 ng/mL in the initial phase and 281 ± 294 ng/mL in the final phase (*p* < 0.001). Group 2 had mean serum ferritin levels of 1216 ± 631 ng/mL in the initial phase and 243 ± 184 ng/mL in the final phase (*p* < 0.001). In addition, Figure 1 shows no significant differences for mean serum ferritin values between groups 1 and 2 in the final phase (*p* = 0.53). Additional file 2: Appendix A shows that there was no correlation between serum ferritin (initial and final) values and SF-36 domains, except a weak correlation between physical functions and serum ferritin in the final phase (r = −0.352, *p* = 0.008).

Table 2 shows the average values of the SF-36 domains related to patient groups. Six domains were significantly different between groups 1 and 2: bodily pain (*p* = 0.03), general health perception (*p* = 0.03), vitality (*p* = 0.01), social functioning (*p* = 0.01), role emotional (*p* = 0.001) and mental health (*p* = 0.04). Group 1 demonstrated lower average values for these six domains when compared with group 2. In addition, patients with *HFE* p.Cys282Tyr homozygous genotype, who had four SF-36 domains worse in the initial phase of this study, remained worse, and further, had two worsening domains (role emotional and mental health) in this final phase. This finding was observed even with the average treatment duration of 3 years and despite a similar decrease in ferritin values to that of group 2, reaffirming the impact of the identified genotype on the QL of hemochromatosis patients.

## 4. Discussion

Some studies reported interesting data on QL in patients with hemochromatosis; however, to the best of our knowledge, the present study is the first to evaluate SF-36 over different periods of treatment. Graaff et al. conducted a study using the Assessment of Quality of Life 4D instrument (AQOL-4D) in 270 patients with a hemochromatosis diagnosis in Australia, assessing the usefulness of the population’s health status. The range of the tool was from −0.04 to 1.00, where a score of 1 represents optimal health, a score of 0 represents a state equivalent to death, and a score below 0 represents a state worse than death. The average usefulness for the general population was 0.81, and that of patients with hemochromatosis was 0.66. The symptomatic states of hemochromatosis presented less utility with respect to the asymptomatic states of the disease [19]. Some studies did not identify significant differences in the QL between hemochromatosis patients and subjects without iron overload. Shaheen et al. conducted a study in the United States with 126 hemochromatosis patients with *HFE* homozygosity for the p.Cys282Tyr or compound heterozygosity for the p.Cys282Tyr/p.His63Asp mutation subjects, but who had evidence of final organ injury or symptoms attributable to iron overload and compared them with their unaffected siblings with the illness. The QL measured by the SF-36 questionnaire between the subjects with hemochromatosis and the unaffected siblings or the population means were not significantly different [20]. Meiser et al. also compared in Australia the QL of 30 patients clinically affected with hemochromatosis and 66 unaffected subjects despite carrying the homozygous genotype of hemochromatosis and did not find any differences between the two groups for both the mental and physical components of the QL questionnaire.

In the present study, there was a greater number of male patients in group 2 when compared with group 1, as shown in Table 1, so much so that there was a significant difference between the groups shown by the *p*-value of 0.001.

It may be related to women’s well-known body protection due to menstruations and pregnancies and, in addition, to the very low penetrance of genotypes from group 2 [5,21]. We observed the worsening of the emotional role domain in group 1, which may have been influenced by the knowledge of a positive result for a genetic test plus the influence of treatment by phlebotomies [22,23,24,25,26]. Rombout et al. reported that 52% of 46 patients during the induction phase of treatment and 37% in the maintenance phase had negative experiences related to treatment, and 16% of patients would even decide not to continue with phlebotomies where alternative options are available [21,26]. Meiser et al. also reported that some patients treated with phlebotomy for a year suffered a decrease in the values of emotional and mental components at this time [20]. Brissot et al. also reported the incidence of the frequency of phlebotomies in the daily life of patients who received phlebotomies once a week; 26% (*n* = 23/90) expressed feeling uncomfortable or very uncomfortable with previous procedures, during and after the bleeding. Many of them expressed a definite desire to stop phlebotomy if there was another treatment available due to all the discomfort that this procedure represented for them [27]. Here, it is important to highlight the necessity to observe the emotional and mental aspects of hemochromatosis patients and to explain that treatment is the best option for their health and QL.

Comparing the initial and final ferritin values of the studied groups, although the values decreased significantly, we observed that they did not reach the proposed objectives by the current international recommendations for the treatment of hemochromatosis, namely reaching and maintaining serum ferritin levels around 50 ng/mL [8]. American and European guidelines also recommend serum ferritin values at 50–100 ng/mL in the maintenance phase [28,29,30]. Thus, we were not able to identify a possible relationship between QL and values of serum ferritin ~50 ng/mL.

The differences in the domains’ values between the two genotypic groups can be explained, at least in part, by the genetic basis of the disease and its respective mutations, iron status, and phlebotomies, even we were not able to identify a significant association with some of these variables in the present study. The EMQN guide detailed the respective associations between genotypes and clinical penetrance with respect to possible ferritin and transferrin saturation values in patients [31]: p.Cys is compatible with hemochromatosis related to *HFE* in the presence of evidence of iron overload; patients with compound p.Cys282Tyr/p.His63Asp may be in danger of developing mild to moderate iron overload in association with comorbidity factors (metabolic syndrome or chronic alcoholism); otherwise, carriers of the p.Cys282Tyr changes in heterozygosis, homozygosis for p.His63Asp, and heterozygosis for p.His63Asp have no greater risk of developing hemochromatosis related to *HFE*.

The worsening in quality of life, even with treatment, can be associated with other pre-existing diseases. According to the therapeutic recommendation for hemochromatosis, the ideal would be to maintain treatment and follow-up because, without this follow-up, the quality of life could worsen. Recent studies observed a beneficial effect of early and sustained management of patients with iron excess, even when iron load is mild or moderately elevated serum ferritin [8].

### Strengths and Limitations

Our study has some limitations. First, we did not perform a correction for multiple testing. However, the statistical comparison performed was able to capture many SF-36 domains differently with the inclusion criteria for the genotypic groups; furthermore, we identified a numerical difference among groups, which supports these findings (especially in Table 2). Second, liver magnetic resonance imaging was not available in the initial and final phases. Third, during the treatment period, we did not have exact data on some items that could have influenced serum ferritin values, such as inflammatory and/or metabolic conditions/diseases, liver injuries, and/or alcohol consumption. However, it is argued that the initial inclusion criteria were stringent, including transferrin saturation ≥45% and serum ferritin ≥300 ng/mL for males or ≥200 ng/mL for females. Further, strict exclusion criteria were enforced for patients with positive serology for hepatitis, alcoholic liver disease, alcohol consumption (more than 20 g/day), hemolytic anemias, repeated blood transfusions, metabolic syndrome, or insulin resistance not resulting from hemochromatosis. Fourth, the number of phlebotomies was not available for all patients in a network system.

## 5. Conclusions

In this final phase, six SF-36 domains were different between genotypic groups (role emotional and mental health, in addition to the four of the initial phase), reaffirming the impact of identified genotype on the QL of hemochromatosis patients. Furthermore, despite a similar ferritin decrease between both patient groups, no differences were observed for the SF-36 domains.

## Figures and Tables

**Figure 1 genes-13-00118-f001:**
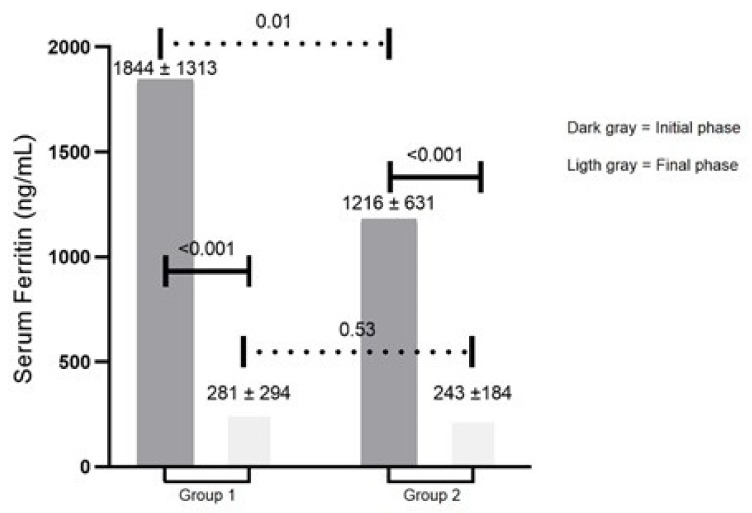
Comparisons of serum ferritin means of groups 1 and 2 in the initial and final phases. Group 1: 23 patients with primary iron overload and homozygosity for the p.Cys282Tyr mutation. Group 2: 42 patients with primary iron overload and other genotypes: compound heterozygosity for the p.Cys282Tyr/p.His63Asp (*n* = 10), heterozygosity for the p.Cys282Tyr (*n* = 4), homozygosity (*n* = 10) or heterozygosity (*n* = 8) for the p.His63Asp, or absence of p.Cys282Tyr and p.His63Asp (*n* = 10).

**Table 1 genes-13-00118-t001:** General characteristics of follow-up according to patient genotypic groups.

	Group 1 ^a^ *n* = 23	Group 2 ^b^ *n* = 42	*p*-Value
Gender (male), %	34.8	85.7	0.001
Age (years), mean ± SD	50 ± 13	54 ± 12	0.21
Serum ferritin (ng/mL), mean ± SD	281 ± 294	236 ± 174	0.53
Number of phlebotomies	8.8 ± 6.3	5.3 ± 5.8	0.72
Self-declared race/color, %			
White	78.3	92.9	0.15
Intermediate	17.4	4.8	
Asian and Amerindian	4.3	2.3	
Level of education, %			
University	65.2	73.8	0.89
Others	34.8	26.2	
Consumption of alcoholic beverages, %			
Never	52.2	31.0	0.20
Occasionally	43.5	57.1	
Frequently	4.3	11.9	

^a^ Group 1: 23 patients with primary iron overload and homozygosity for the p.Cys282Tyr mutation. ^b^ Group 2: 42 patients with primary iron overload and other genotypes: compound heterozygosity for the p.Cys282Tyr/p.His63Asp (*n* = 10), heterozygosity for the p.Cys282Tyr (*n* = 4), homozygosity (*n* = 10) or heterozygosity (*n* = 8) for the p.His63Asp, or absence of p.Cys282Tyr and p.His63Asp (*n* = 10).

**Table 2 genes-13-00118-t002:** Mean (± standard deviation) values of the SF-36 domains according to genotypic groups of the patients.

SF-36 Domains Group 1a	Group 1 ^a^ *n* = 23	Group 2 ^b^ *n* = 42	*p*-Value
Physical functions	84 ± 24	82 ± 22	0.34
Role physical	74 ± 43	73 ± 36	0.13
Bodily pain	64 ± 30	74 ± 24	0.03
General health perception	58 ± 21	69 ± 21	0.03
Vitality	56 ± 24	67 ± 21	0.01
Social functioning	68 ± 31	87 ± 22	0.01
Role emotional	57 ± 45	83 ± 30	0.01
Mental health	69 ± 22	77 ± 20	0.04

^a^ Group 1: 23 patients with primary iron overload and homozygosity for the p.Cys282Tyr mutation. ^b^ Group 2: 42 patients with primary iron overload and other genotypes: compound heterozygosity for the p.Cys282Tyr/p.His63Asp (*n* = 10), heterozygosity for the p.Cys282Tyr (*n* = 4), homozygosity (*n* = 10) or heterozygosity (*n* = 8) for the p.His63Asp, or absence of p.Cys282Tyr and p.His63Asp (*n* = 10).

## Data Availability

We do not wish to share our data because more clinical variables will be studied.

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
