# Peer review of "Quality of Life Scores Remained Different among the Genotypic Groups of Patients with Suspected Hemochromatosis, Even after Treatment Period"

_genes, 2022, doi:10.3390/genes13010118_

Round 1

Reviewer 1 Report

Line 24 ,,,, remove the comma after 'scores'

Line 25 …. place a comma after '3 years'.

Line 35 …. rephrase to ' despite that both patient groups demonstrated similar and significant decreases in serum ferritin values, ….. '

Lines 69 …... statement appears to be incomplete

Line 72... delete 'form'

Line 87... delete 'up'

Line 88, … delete 'a' before guidance

Lines 88 and 89 …. delete 'that was proposed by us'

Line 92... rephrase to 'Thirteen patients were lost to follow-up and one patient died.'

Lines 96-98. These two sentence are quite confusing. What was 'random'? What is 'generally accepted'. This requires clarification.

Line 175 …. rephrase to '…. , remained worse, and further, had two worsening domains........'

Line 183. … delete 'the' before 'treatment'

Line 228.... rephrase, ' reported that some patients treated.....'

Line 233 …. remove 'expressed, ' after definite

Line 238 …. insert a space after 'recommended.'

Lines 247 and 248....... This sentence is quite confusing. Please clarify, especially for those of us that are unfamiliar with how 'Eva está brincandos' pertains.

Line 260... change 'according with' to 'among'.

Line 265... Change to 'However, it is argued that the initial inclusion criteria were stringent, including transferrin saturation.... etc. '

Line 267... Change to 'Further, strict exclusion criteria were enforced for patients with ….. etc'.

Line 272... insert 'SF-36' between 'six' and 'domains'

Reviewer 2 Report

Interesting topic, QoL studies are important. In particular when focussing on the effects of treatment.
In this light it would be recommendable to discuss and speculate on why treatment did not alter QoL (should it have?), and what you would to recommend to clinicians based on these results.

Some more specific suggestions (based on line number provided in the initial manuscript):
Line 45: do you mean the most common genetic mutation?
Line 47: MRI not for diagnosis, but for degree of iron load.
Line 53: Consider to delete this sentence (what is the message you would like to address?).
Add flow chart to explain sentences 88-95. This should be part of the results section.
Line 97: ‘they were randomly included’ what do you mean by this?
Line 122-131: could be combined with the previous mentioned section (88-9%) in the results section.
Table 1: please do not report p-values in a baseline table, or in the text discussing the baseline characteristics.
Please discuss the provided treatment in the text. Phlebotomies? Or were there also patients treated with iron chelating agents?
Did patients stop treatment due to side effects, or were treated suboptimal (see ferritin values) due to side effects?
Delete line 173.
In the first paragraph of the discussion you summarize results that were not previously discussed. Please add these results to the ‘results’ section.
It may be insightful to compare results to a control population of healthy volunteers (or previously published results from a healthy population.
Line 180-218: please summarize only relevant previous findings and highlight the differences with your study. Relevant in this case means studies that also specifically looked at treated patients (with near-normal ferritin levels). Note if their findings are in line with your findings, or how differences may be explained.
Line 248: my Portugees is not that well, but google translate did the job.

Overall, the discussion is too long. Particularly the part summarizing all previous work. Please confine the discussion (section 4.1) to the following:

  • How can the (lack of) differences in QL before and after be explained (and what could we have expected);

  • How, in you opinion, could the difference between HFE and other mutations be explained.

Focus in the strengths and limitations on the most important points and elaborate on them.

Round 2

Reviewer 2 Report

Improved version upon revision.